

# A Double ITCZ Phenomenology of Wind Errors in the Equatorial Atlantic in Seasonal Forecasts with ECMWF Models

Jonathan K. P. Shonk[1], Teferi D. Demissie[2], Thomas Toniazzo[2]

[1] National Centre for Atmospheric Science, University of Reading, Reading, UK

[2] Uni Research, Bjerknes Centre for Climate Research, Bergen, Norway

*Correspondence to*: Jonathan K. P. Shonk (j.k.p.shonk@reading.ac.uk)

**Abstract.** Modern coupled general circulation models produce systematic biases in the tropical Atlantic that hamper the reliability of long-range predictions. This study focuses on a common springtime westerly wind bias in the equatorial Atlantic in seasonal hindcasts from two coupled models – ECMWF System 4 and EC-Earth v2.3 – and in hindcasts also based on

System 4, but with prescribed sea-surface temperatures. The coupled models share common atmosphere and ocean components, although at different versions.

We examine the sequence in which different biases appear during the development of the westerly bias in early April, which is marked by a rapid transition from a wintertime bias pattern with an equatorial cold tongue and an easterly wind bias to a springtime westerly bias regime displaying a marked double ITCZ. The transition is a seasonal feature of the model climatology

(independent of start date), and is associated with the seasonal increase in rainfall around the start of April and the consequent enhancement of the southern branch of a double ITCZ, which generates excess off-equatorial convergence and redirects the trade winds away from the equator.

There is no evidence of remote influences on the biases at the time of the transition. By contrast, there appears to be an association with a persistent dry bias north of the equator. Based on our analysis, a possible contribution to the springtime

development of the double ITCZ and the westerly equatorial wind bias is a failure to correctly represent the meridional cross-equatorial flow, which can instigate a meridional rainfall bias pattern across the equator.

## 1  Introduction

The identification and reduction of systematic biases in coupled general circulation models (CGCMs) has been an ongoing problem in model development in recent times. Despite significant development of CGCMs in the last two decades, large

systematic biases remain in the simulated tropical climate – including the tropical Atlantic (Solomon *et al*, 2007; Davey *et al*, 2002; Richter and Xie, 2008; Toniazzo and Woolnough, 2014) – and these biases can have significant impacts on seasonal forecasts and future climate predictions. A common problem is the misrepresentation of the direction of the annual mean zonal gradient of sea surface temperature (SST) over the equatorial Atlantic, with a tendency for models to simulate an eastward gradient with colder SSTs in the west and warmer SSTs in the east (Davey *et al*, 2002; Klein *et al*, 2013; Richter *et al*, 2014).



This reversed gradient is at least partly due to the failure of CGCMs to reproduce the observed cold tongue (a region of colder water extending along the equatorial Atlantic from the east) and the associated shallow mixed layer in the eastern equatorial Atlantic during boreal summer (Davey *et al*, 2002; Richter and Xie, 2008). In turn, this has been shown to be associated with, and in some cases the result of, a westerly equatorial surface wind bias that develops during boreal spring (Chang *et al*, 2007; Richter *et al*, 2012, 2014). The erroneous seasonal slackening of the equatorial easterlies also has significant implications for the coastal climate of southwest Africa, where a systematic warm bias develops in CGCMs partly in response to wind-forced equatorial thermocline anomalies that propagate poleward along the coast (Toniazzo and Woolnough, 2014; Voldoire *et al*, 2014; Voldoire *et al*, 2018).

A number of hypotheses have been made concerning the possible root causes of the springtime surface westerly wind bias over the equatorial Atlantic. While it is most pronounced in CGCM simulations, an equatorial westerly bias can also be identified in atmospheric GCM (AGCM) simulations with prescribed SSTs (Richter and Xie, 2008), indicating that errors in both the atmosphere and ocean components of a CGCM could be responsible. Chang *et al* (2008), Wahl *et al* (2011) and Richter *et al* (2012) showed that the westerly bias is associated with biases in sea-level pressure gradient in the atmosphere component, rainfall biases in the Amazon and West Africa, and erroneous boundary layer entrainment. However, a direct causal link between these concomitant model biases was not clearly established. Studies have also suggested contributions from the ocean component, such as erroneous stratification and insufficient upwelling in the cold tongue (Breugem *et al*, 2008; Exarchou *et al*, 2018) and poorly resolved dynamical features associated with a model grid that is too coarse (Seo *et al*, 2006). It is important to identify the most prevalent causes of the development of the westerly bias, since it is linked with a failure to correctly simulate the mean seasonal cycle of equatorial winds and SST, which affects the ability of CGCMs to predict tropical Atlantic variability such as the Atlantic Niño. Also, as a result of the warm SST biases on the equator and along the coast, the simulation of the present and future climate may be regarded as unreliable, especially if employed to forecast rainfall over decadal and seasonal time scales in the tropical Atlantic and the adjoining continental monsoon areas (Hulme *et al*, 2001). Linking systematic biases with process errors in the models can help not only to improve their performance but also the initialisation procedures for seasonal to decadal forecast applications, which would ultimately lead to a more skilful forecast over the tropical Atlantic.

In this study, we analyse systematic biases in the tropical Atlantic in seasonal hindcasts (also sometimes referred to as reforecasts) obtained with the System 4 coupled seasonal prediction system of the European Centre for Medium-Range Weather Forecasts (ECMWF). These are compared with hindcasts obtained by prescribing the time-dependent SST fields from observations in order to isolate biases originating from the atmosphere component of the model and those originating from air–sea coupling. We also compare the results with hindcasts based on the coupled EC-Earth v2.3 model in order to assess the dependence of biases over the equatorial Atlantic on model physics and on biases occurring elsewhere. The analysis methodology is similar to that adopted by Shonk *et al* (2018) and Toniazzo and Woolnough (2014), in that clues on the causal links between different biases are derived from the temporal sequence with which they appear in the course of the hindcasts, as well as on the relationship, or lack thereof, with more or less evolved SST biases. Using this methodology, we probe the



origins of the springtime westerly zonal wind bias in an attempt to build an understanding of the processes that lead to its development and ultimately hypothesise where its root cause may lie. In the next section of this paper we introduce the models in more detail and describe the datasets. In Section 3, we describe the evolution of systematic hindcast biases in the Atlantic basin. In Section 4, we investigate relationship of such biases with biases that develop outside the tropical Atlantic. In Section 5, we discuss some hypotheses of bias origins based on the results from the previous two sections. We conclude the paper in Section 6.

## 2  Approach and Method

We use hindcast data from two models in this study. The first is ECMWF System 4 (referred to as "S4" in this paper), a model designed for operational seasonal forecasting; the second is EC-Earth version 2.3 ("EC"), a model designed as a tool for climate research. S4 combines cycle 36R4 of the Integrated Forecast System (IFS; Molteni *et al*, 2011) with version 3.0 of the Nucleus for European Modelling of the Ocean (NEMO; Madec, 2008), coupled using a version of the Ocean Atmosphere Sea Ice Soil 3 coupler (OASIS3; Valcke, 2013). The IFS's dynamical core uses a spectral horizontal discretisation and hybrid sigma-$p$ levels in the vertical. Cycle 36R4 has a quadratic truncation at wavenumber 255 on 91 vertical levels that extend from the surface to 0.01 hPa. The physics calculations are made on a Gaussian N128 grid, giving a horizontal resolution of about 0.7°. The NEMO ocean component uses a finite-difference discretisation on the curvilinear ORCA1 grid, which has a horizontal resolution of about 1° (with increased meridional resolution at the equator) and 42 levels in the vertical.

The EC model also uses the IFS atmosphere model, but cycle 31R1 (the same version on which the ERA-Interim reanalysis product is based). It is truncated at wavenumber 159 and has 62 levels up to 5 hPa; the physical parameterisations are computed on a reduced Gaussian N80 grid, corresponding to a horizontal resolution of 1.125°. EC uses version 2.0 of NEMO as its ocean component, which runs on the same ORCA1 grid as the version of NEMO in S4, and OASIS3 for coupling. Table 1 presents a summary of the differences between the two models. Full details on S4 and EC are presented by Molteni *et al* (2011) and Hazeleger *et al* (2010, 2012) respectively.

Operational hindcasts are available from S4 starting on the first of every month and run for at least seven months; hindcasts are available from EC starting on 1 February, 1 May, 1 August and 1 November, and run for four months. A further set of hindcasts is available from an "atmosphere-only" version of S4, which we refer to as "S4A". This uses the same version of the IFS as S4, but with SST prescribed from the OISST dataset (Reynolds *et al*, 2002). The S4A hindcasts are initialised on the same four dates in the year as the EC hindcasts, and are also run for four months.

For the hindcast climatologies of S4 and S4A, we select a subset of 14 years, spanning 1996 to 2009; for EC, we select a 20-year period spanning 1981 to 2000. During these periods, both S4 and EC are initialised with atmosphere data from ERA-Interim (Dee *et al*, 2011) and ocean data from ORA-S4 (Balmaseda *et al*, 2013). We preclude years more recent than 2009 in S4 and S4A as, from 2010 onwards, initial conditions are obtained from operational analyses instead of ERA-Interim. For EC, we preclude years after 2000 as, from 2001 onwards, there was a change in the initialisation method. Hence, for all models,



these ranges represent the most up-to-date range of years available with consistent initialisation data source and method. For S4 and S4A, we extract eight ensemble members for each start date, as this was found to be sufficient to represent model uncertainty. For EC, we use all ten ensemble members that are available. Ensemble means are used throughout to represent the models' best estimates of the conditions.

**Table 1.** Summary of the main features of the two coupled models used in this study. The features of uncoupled model S4A are the same as S4, but with the SST prescribed and hence no ocean component.

|  | ECMWF System 4 (S4) | EC-Earth v2.3 (EC) |
|---|---|---|
| *Main purpose* | Seasonal forecasting | Climate studies |
| *Atmosphere component* | IFS, cycle 36R4 | IFS, cycle 31R1 |
| *Horizontal grid (resolution)* | TL255/N128 (~0.7°) | TL159/N80 (~1.125°) |
| *Vertical levels* | 91 | 62 |
| *Ocean component* | NEMO, version 3.0 | NEMO, version 2.0 |
| *Horizontal grid (resolution)* | ORCA1 (~1° *) | ORCA1 (~1° *) |
| *Vertical levels* | 42 | 42 |
| *Coupling component* | OASIS3 | OASIS3 |
| *Time Step* | 45 min | 45 min |
| *Coupling Interval* | 3 hours | 3 hours |

* Increased meridional resolution at the equator – up to about ⅓°.

Our observation datasets have been selected to match the initialisation datasets where possible. We take SST observations from OISST and wind observations from ERA-Interim. Observed rainfall is taken from the Tropical Rainfall Measuring Mission (TRMM; Kummerow *et al*, 2000). As in Toniazzo and Woolnough (2014) and Shonk *et al* (2018), we base the analysis of systematic biases and their evolution on the comparison between climatologies from the hindcasts (one daily or monthly mean climatology for each lead time in the hindcast) with a corresponding daily or monthly mean climatology from the observations. Where possible, these climatologies have been constructed from matching subsets of years (validity time in the hindcasts) in order to minimise bias contributions originating from observed and simulated interannual variability. However, when the overlap between years in different datasets is limited, we deviate from this rule. In any case, in this analysis we focus





on systematic biases only, and we have ensured that none of the results presented here is sensitive to details of the time period chosen to define the climatologies. Where statistical significance is calculated for a certain mean bias, this is obtained by considering the statistical distribution of that bias in the available ensemble of hindcasts.

## 3 Model Biases in the Tropical Atlantic

### 3.1 Annual Cycle of "Climatological" Biases

The biases in the seasonal climatology of S4 hindcasts, defined as the average bias in the seventh month of hindcast for each start date, are shown in Figure 1. For the rest of this paper, season names are defined in terms of the northern hemisphere for brevity. The rainfall bias shows a consistent pattern throughout the year over the tropical Atlantic: a dry band north of the equator and a wet band to the south. The pattern stretches zonally across the central Atlantic and is consistent with the presence of a double Intertropical Convergence Zone (ITCZ), with an apparent second branch south of the equator. The magnitude of the bias pattern varies over the course of the year, with the largest biases occurring in winter and spring. In the west, we see contrasts in bias off the coast of South America, where there is tendency for a wet bias over the sea and a dry bias over the adjacent land, the latter of which is particularly marked in spring and summer. S4 also has a warm SST bias off the coast of southwest Africa, which persists all year. In summer, this warm bias extends into the region of the Atlantic cold tongue; in autumn, this is replaced by a cold bias extending along the equator from the west.

There is also a seasonal cycle in the wind biases in the tropical Atlantic. The climatological winds here are easterly trade winds, and the S4 biases in autumn and winter are easterly, implying a strengthening of the trade winds. In spring, however, the bias in zonal wind turns westerly, implying a weakening of the trade winds. This "flip" of the climatological wind bias affects much of the tropical Atlantic, with the westerly bias extending from the coast of South America to the edge of the Gulf of Guinea. In summer, the westerly bias weakens with a strong convergent flow into the region of excess rainfall on the equator. The patterns of bias identified in S4 reflect patterns found in many coupled models in the tropical Atlantic (Huang *et al*, 2007; Richter and Xie, 2008).

Figure 2 shows the seventh-month biases in the equatorial Atlantic from month to month in more detail. Given the tendency of the bias patterns in the tropical Atlantic to be roughly zonal in structure outside the Gulf of Guinea and away from the coast of South America, the zonal mean biases are calculated in the longitude range 40° W—0°. The transition of zonal wind bias from easterly to westerly takes place between February and April (Figure 2c). The westerly bias dominates the tropical Atlantic through April and May before drifting northwards and weakening over the subsequent few months. Eventually, it is replaced by an easterly bias that persists for the rest of the year with varying magnitude.

An increase in rainfall bias over the tropical Atlantic occurs in the months following the onset of the westerly wind bias. Figure 2b indicates a strengthening of wet bias to the south of the equator in April, which grows in both intensity and meridional extent to cover much of the tropical Atlantic in May and June. The result of this strengthening is an overall wet bias in rainfall




in the region 20° S—20° N and 40° W—0° (Figure 2a). At the same time, as the equatorial wind bias becomes westerly, the cold SST bias that persists throughout most of the rest of the year (indicative of a cold tongue that is too strong) fades and eventually turns warm in April and May (Figure 2d). In the summer, the rainfall biases decrease and the warm SST bias reverts to cold. Additionally, we see significant SST biases in the subtropics, with a cold SST bias south of the equator that persists

5   all year and seasonally varying biases to the north that switch between warm and cold.

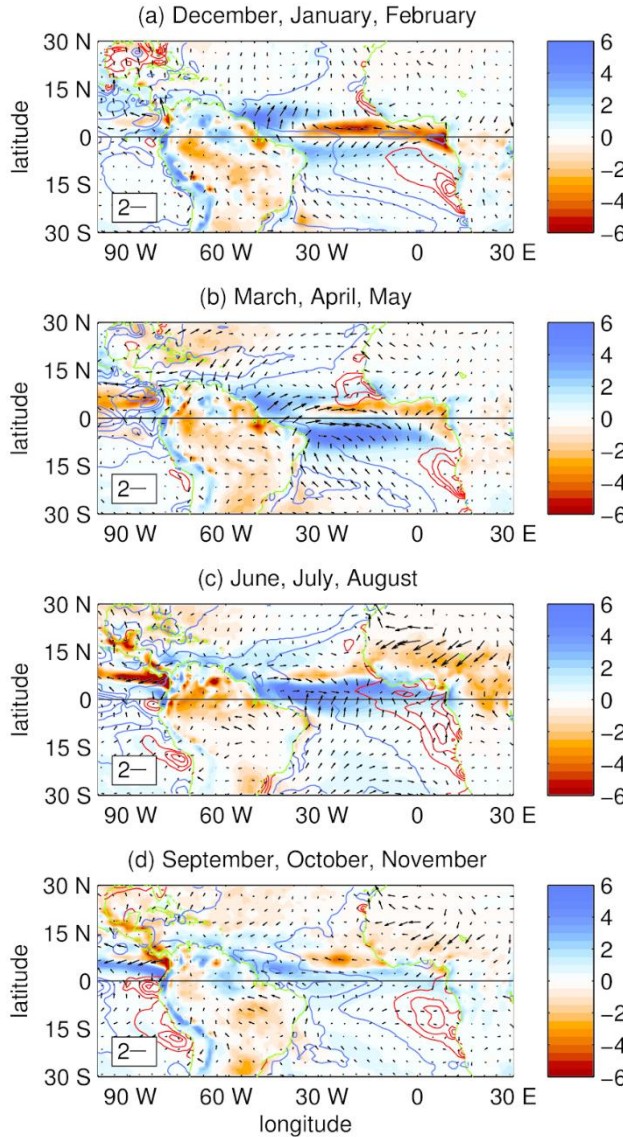

**Figure 1. Maps of seasonal mean biases in rainfall (filled contours), sea-surface temperature (open contours) and 10 m wind vector. Rainfall is in mm d⁻¹ (see colour bar); wind vector is in m s⁻¹ (see legend for scale), and sea-surface temperature contours are in 0.5 °C increments each side of zero – red for warm biases, blue for cold biases. Biases are calculated for the seven-month hindcasts and averaged over the seasons, and over years 1998 to 2009 for coupled model S4.**



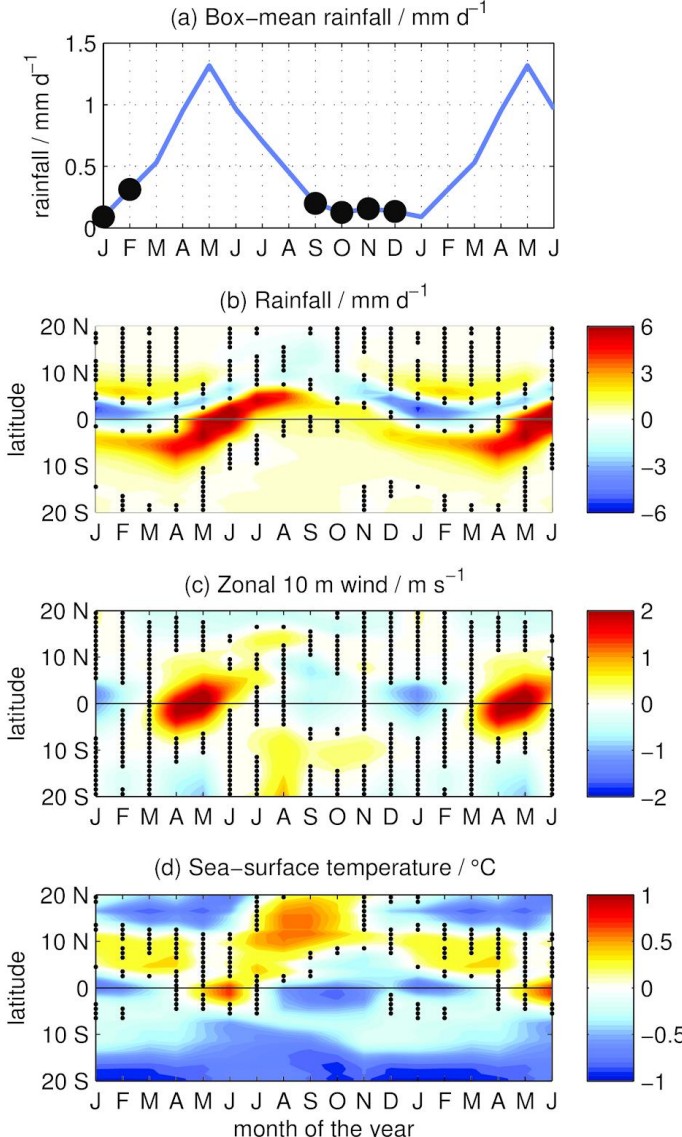

**Figure 2. Latitude–time plots of zonal mean biases in (b) rainfall, (c) zonal wind at 10 m and (d) sea-surface temperature. Biases are calculated for the seventh month of hindcast and averaged zonally over longitudes 40° W—0°. Panel (a) shows the bias in box-mean rainfall after seven months in the same longitude range, and latitudes 20° N—20° S. Insignificant regions of the plot with respect to interannual variability are indicated with black spots.**

The set of phenomena described here suggests that the biases that develop in S4 during spring are of a particular nature and different to those that occur throughout the rest of the year. As stated in the introduction, the importance of biases in spring for the evolution of the simulated climatology later in the year has been often noted in the literature, with a strong equatorial westerly bias known to affect the simulated climatology along the African coast. Understanding the mechanisms that cause



these spring bias patterns to develop is an important step in improving our seasonal forecast skill at this crucial time of year (e.g. Voldoire *et al,* 2018). We therefore focus our attention on spring.

## 3.2  Linking the Biases in the Three Models

To examine the development of the equatorial bias and its relationship with biases in other aspects of the atmospheric
circulation, we consider its evolution at sub-synoptic time scales. We compare the bias evolution between different models to understand common behaviours and identify the impacts of biases in SST.

We begin by examining zonal mean quantities averaged across the central Atlantic as in Figure 2 (longitude range 40° W—0°) but for daily-average climatologies, focussing on the first 120 days of the hindcasts initialised on 1 February. Figure 3a shows the evolution of observed zonal-mean rainfall and wind vector at 10 m for this period, during which the core of the
ITCZ drifts northwards from 1° N to 5° N. Between March and April there is a widening of the rainfall band, which extends to the south of the equator. The winds maintain an easterly zonal component, and a meridional component that is convergent towards the ITCZ.

The rainfall bias pattern in S4 seen in Figure 1 does indeed correspond to the development of a double ITCZ structure in the hindcasts (Figure 3b). There is evidence of its development as early as mid-February, although it becomes much clearer in
March and April. The northern ITCZ is displaced slightly to the north of its observed position and is markedly weaker, while the erroneous southern ITCZ develops south of the equator, producing the strong wet bias seen in Figures 1 and 2. There is a marked strengthening of the rainfall in both ITCZ bands around the start of April that persists into May, although the double ITCZ structure becomes less discernible by the end of May. The cold SST bias on the equator also begins to develop in mid-February, growing in late February and March in the same location as the dry region which separates the two branches of the
ITCZ. The transition from a cold SST bias on the equator to a warm bias then occurs in late March and early April, becoming warm by mid-April and warming further in May (Figure 4a).

The initial easterly wind bias on the equator develops within the first ten days of hindcast (before any clear sign of a double ITCZ or a cold SST bias) and persists through February and March (Figure 3b). The change of sign of the zonal wind bias happens around the same time the SST bias starts its transition from cold to warm, and occurs more suddenly than it appears
in Figure 2. By day 50, the easterly bias has faded to near zero. In early April, the westerly biases start to grow rapidly and persist through April and May. The magnitude of this bias indicates that the trade winds on the equator reduce to near calm conditions. The rapid westerly bias growth is found to occur consistently in early April when averaging over subsets of hindcast years.





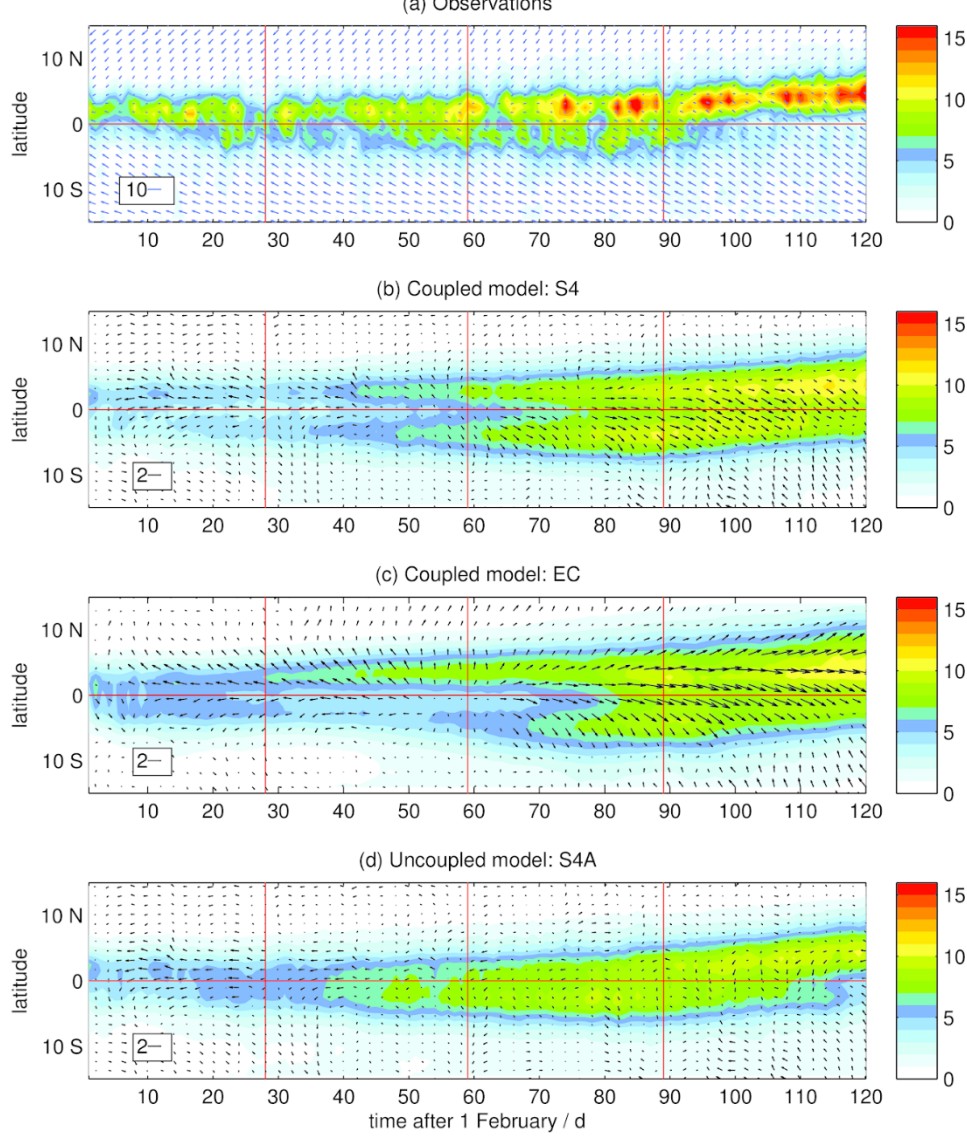

**Figure 3.** Latitude–time plot showing the evolution of model biases, all averaged over longitudes 40° W—0°. Panel (a) shows observed rainfall as filled contours, with observed mean vector wind at 10 m (blue arrows). Panels (b) to (d) show the rainfall according to the three models, with the wind vector biases included as black arrows. Note that the wind vectors point in the direction they would if they were on a map; the horizontal dimension here is time rather than distance. Red lines show the boundaries between months and the equator. Rainfall is in mm d$^{-1}$ (see colour bar); wind is in m s$^{-1}$ (see legend). Averaged over years 1996 to 2009 (S4 and S4A) and years 1981 to 2000 (EC). TRMM rainfall data in panel (a) is averaged over years 1998 to 2009.

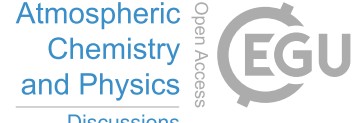



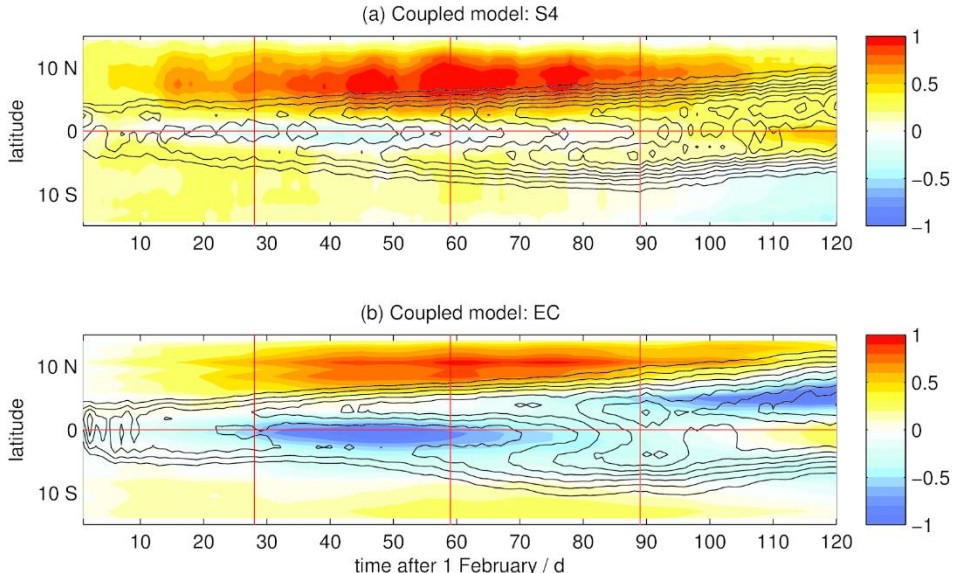

**Figure 4. Latitude–time plot, averaged as in Figure 3, showing biases in sea-surface temperature in the coupled models (filled contours). The rainfall values from the models are shown here as black contours with 1 mm d⁻¹ interval, starting from 3 mm d⁻¹. Sea-surface temperature biases are in °C (see colour bar).**

Given the sudden nature of the onset of westerly bias, we define two separate bias regimes that occur in S4 in spring: an easterly bias regime in February and March, associated with cooling SSTs and a strong double ITCZ; and a westerly bias regime in April and May, associated with (gradually) warming SSTs and a weakening double ITCZ. Figure 5 probes the timing of this transition using hindcasts initialised at the start of March, April and May. The easterly bias regime is evident at the start

of both the February and March hindcasts, although in the latter the easterly wind bias is much weaker and constrained to a narrow band north of the equator. Accordingly, the cold SST bias is weaker, implying that the development of biases in the easterly regime has some dependence on start date and lead time. The onset of the westerly bias regime, however, occurs consistently at the start of April in the February, March and April hindcasts, implying that the transition is a seasonal feature of the climatology of S4. Westerly biases also develop within the first few days of hindcasts initialised after the transition, as

seen in the May hindcasts.

A comparison with the two other models used in this study, S4A and EC, reveals further clues to the reason for the switch in the equatorial circulation bias. In EC, the general evolution of bias patterns is similar to that in S4 (Figures 3c and 4b), suggesting that relationships between the SST, rainfall and wind fields may be explained by similar mechanisms. Initially, an easterly bias develops in EC on the equator with a developing cold SST bias and a double ITCZ structure, similar to S4. Then,

the easterly wind bias becomes westerly and the cold SST bias reduces and turns warm between April and May. The rainfall in both branches of the ITCZ increases in early April and the double ITCZ structure persists well into May. In other words, there is evidence of the same two-regime bias pattern.



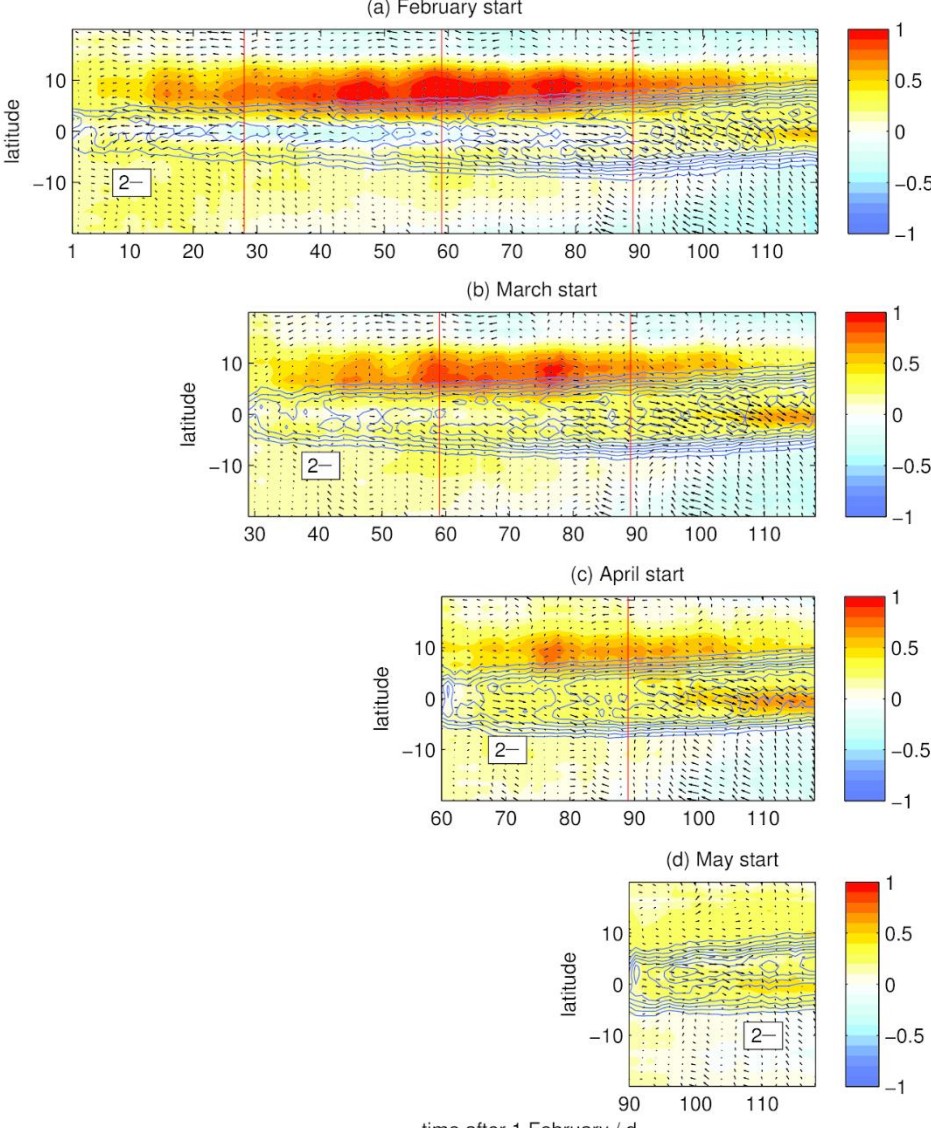

**Figure 5. Staggered latitude–time plot, averaged as in Figure 3, showing biases in sea-surface temperature (filled contours) and wind vector bias at 10 m, with contours of rainfall (in 1 mm d$^{-1}$ increments from 3 mm d$^{-1}$ upwards). Panel (a) shows the bias development in the S4 hindcasts starting in February; panels (b, c, d) show the development in the first days of the March, April and May hindcasts. Sea-surface temperature biases are in °C (see colour bar); winds are in m s$^{-1}$ (see legend).**

The details of the bias patterns show both differences and commonalities. In EC, the patch of cold bias that develops in February and March is stronger and extends further south, suppressing convection south of the equator. The result is that the southern ITCZ is weaker than the northern ITCZ in EC (in contrast to S4, where the southern ITCZ is slightly stronger), and





that a fully discernible double ITCZ structure does not appear until the end of March. But the initial development of the easterly bias is similar in EC and S4, with the appearance of an easterly bias in the first ten days that leads the development of any double ITCZ or cold SST bias, and a transition to westerly bias that also occurs around the start of April despite differences in the distribution of rainfall and SST bias at the time.

Comparing S4 and S4A enables us to examine the effect of coupling on the biases. With zero SST bias effectively prescribed on the equator, there is no suppression of convection over the equator and hence a clear double ITCZ bias pattern is not seen (Figure 3d). However, the model ITCZ is still generally too broad, extends too far into the southern hemisphere, and also increases in strength on both sides of the equator in early April. There is evidence of a weak double ITCZ structure at times, with rainfall maxima generally located either side of the equator rather than on it.



**Figure 6. Latitude—time plot, averaged as in Figure 3, showing the coupling effect on rainfall (filled contours, in mm d⁻¹) and wind vector in m s⁻¹.**

As in S4, the initial easterly wind bias develops within the first ten days – indeed, the difference in wind bias between coupled and uncoupled models in February and March is near zero despite the different rainfall pattern (Figure 6). The full transition into westerly bias, however, does not occur in S4A. The easterly wind bias fades in late March, and the bias generally remains weak through April and May with varying zonal component. In terms of coupling effect (Figure 6), the transition in behaviour between the first two months and the second two months is remarkable, with a strong westerly coupling effect rapidly

developing at the start of April and persisting through April and May, and dominating the tropical Atlantic.

In both the coupled models, the initial bias development in the easterly bias regime shows a classic double ITCZ bias pattern, with excess easterly winds developing first and leading to subsequent cooling and suppression of rainfall over the equator. The lag between wind bias growth and SST bias growth, combined with the similarity of the pattern of easterly bias across all models despite the differences in SST bias pattern, suggest that the wind bias leads the initial development of the double ITCZ

pattern. The similarity of the wind bias growth between S4 and S4A implies an insensitivity to coupling that suggests that the origin of the easterly regime biases lies in the atmosphere component of the models. These results echo those of Shonk *et al* (2018), who found initial easterly wind biases in the atmospheric component of S4 to be pivotal in explaining the root cause of rainfall biases in the western Pacific Ocean.



All three models then show some degree of transition away from the easterly bias regime. The transition can be divided into two parts: first, a weakening of the easterly bias that is common to all three models, followed by a development of strong westerly biases that is common only to the coupled models. The seasonal nature of the transition implies an association with an aspect of the seasonal cycle of the model climatology and, for the initial reduction of the easterly bias the most likely candidate is the systematic increase in rainfall that is common to all three models. The growth of the westerly bias then occurs only to an extent in the coupled models, indicating that the presence of a double ITCZ is important.

It should be noted, however, that the presence of a double ITCZ (at least in S4) is not entirely reliant on the presence of a cold SST bias on the equator. Figure 5 shows a clear double ITCZ structure that develops rapidly in the April hindcasts despite the initial SST bias on the equator being warm, which implies that there must be another mechanism contributing to the development of a double ITCZ, most likely associated with the coupling. We return to this point in Section 5.

### 3.3 Focussing on the Transition Period

From the analysis in the previous subsection, it appears that the "flip" in sign of the zonal wind bias in the equatorial Atlantic at the start of April is associated with a transition between two distinct bias regimes that have different associations with the rainfall. This first consists of an easterly bias, still generally associated with a double ITCZ but not obviously dependent on it; the second consists of a westerly bias that is always associated with a double ITCZ. In the following, we focus on the transition period in more detail and look for further clues in the zonal distributions of bias.

Figure 7 shows model biases around the start of April, comparing dekad-mean rainfall and wind vector for the sixth to ninth dekads (10-day periods) after 1 February, a period spanning the end of March and most of April. The observed ITCZ (Figures 7a, 7e, 7i and 7m) is situated a few degrees north of the equator throughout and, over the ocean, its heaviest rainfall is generally situated over the central Atlantic. A further rainfall maximum is centred on the mouth of the Amazon, with rain falling both onto land and sea. There is also a secondary zonal band of rainfall situated just to the south of the main ITCZ that extends eastward from the coast of Brazil out to a longitude of between 30° W and 20° W, and south of the equator.

In these dekads, two bias patterns affect the Atlantic rainfall patterns in coupled models S4 and EC: the presence of a double ITCZ, and a zonal dipole bias pattern with too much rain in the west and too little in the east. Together, these combine to produce a southern ITCZ with its rainfall maximum locked to the coast of South America. The distribution of the rainfall between the two ITCZ bands is different in S4 and EC, with S4 producing a stronger southern ITCZ and EC producing a stronger northern ITCZ (as seen in the previous subsection). However, in both coupled models, the southern ITCZ grows in strength through these four dekads, particularly in dekads 8 and 9.




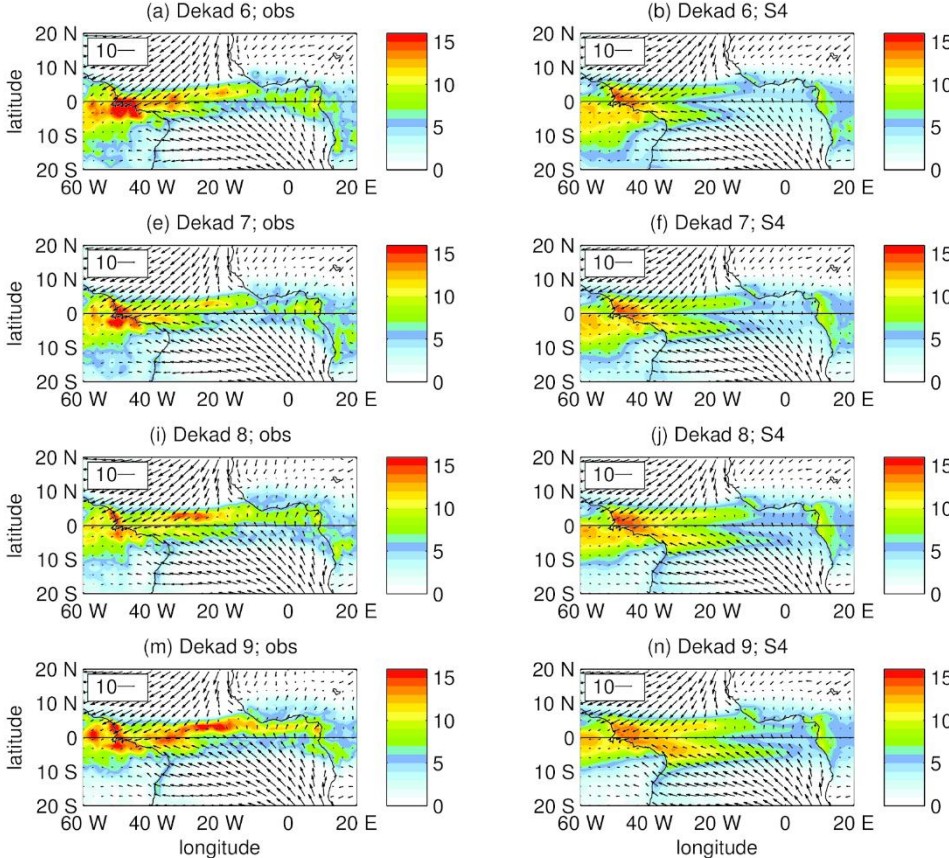

**Figure 7. Maps comparing rainfall and wind at 10 m in the observations and all three models. The observed panels (left column) show dekad-mean rainfall (filled contours, in mm d$^{-1}$) for the sixth to ninth dekads (10-day periods) after 1 February, with wind at 10 m (in m s$^{-1}$) as black arrows. The other three columns show the same for the three models, with data taken from the February hindcasts.**

The result of this strengthening is excess convergence off the coast of South America. Observed winds in the Atlantic consist of south-easterly trade winds across the southern hemisphere and north-easterly trade winds across the northern hemisphere that converge into the ITCZ. In the coupled models, the southern hemisphere trade winds are directed further south into the strengthening southern ITCZ. The location of the northern trade winds, however, remains largely unchanged, leaving a windless strip along the equator that the trade winds do not reach. The strengthening of the rainfall in the southern ITCZ through this period aligns well with the fade of the equatorial winds to zero in S4 and EC (Figure 7), with the onset of the westerly wind bias occurring at about day 60 (the start of April). The consequent redirection of moisture transported by the trade winds into the southern ITCZ is likely to contribute to this strengthening.

In S4A, by contrast, the weak double ITCZ structure is not strong enough to deflect the winds away from the equator. The equatorial winds therefore do not fall to near zero and the full westerly bias does not develop. The rainfall off the coast of South America lacks a strong dry slot, hence convection can remain much closer to the equator and the southern hemisphere



trade winds can reach the equator throughout the period. However, S4A does still produce too much convection south of the equator, which may still contribute to the reduction of the easterly bias at the end of March.

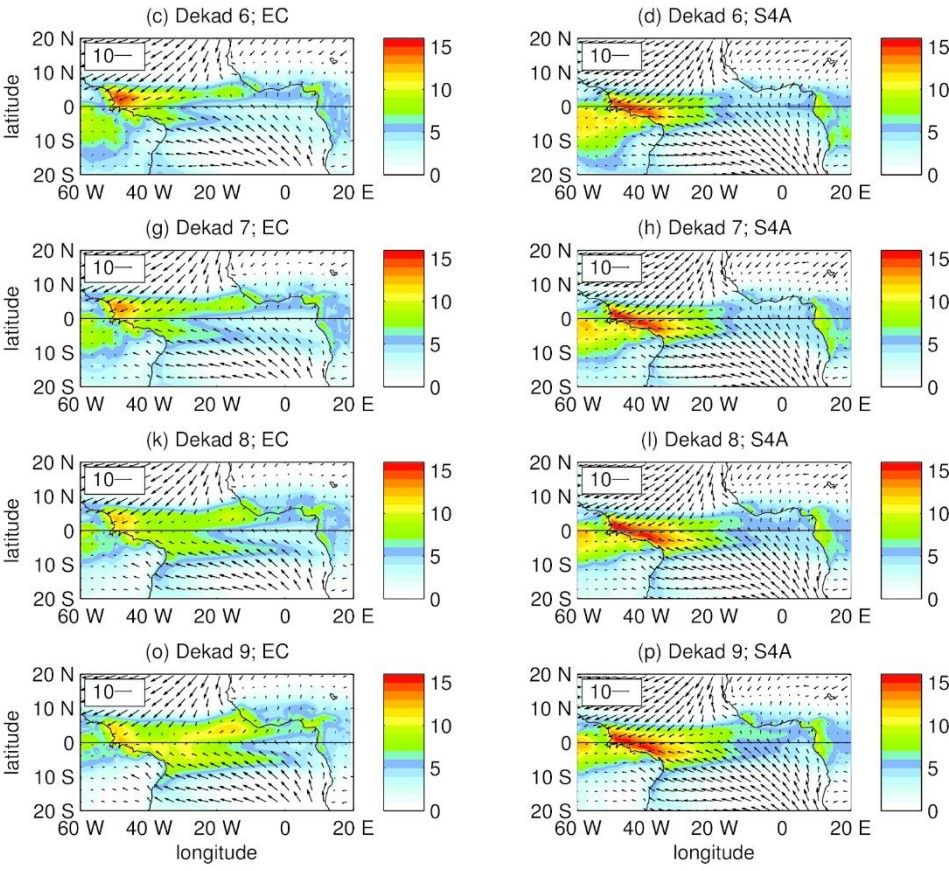

**Figure 7. (Continued)**

In summary, the onset of the westerly bias at the start of April in fact corresponds to a rapid reduction in trade wind strength from stronger than observed to very light. The common enhancement of rainfall around the start of April, in combination with the tendency of all three models to produce too much convection south of the equator in the western Atlantic, generates excess convergence south of the equator and redirects the trade winds. However, as the coupled models show a full double ITCZ, the southern hemisphere trade winds are deflected further from the equator, leaving a zone of near-zero winds along the equator. Hence the systematic increase in rainfall and the emergence of a double ITCZ at the start of April are crucial factors in the onset of the spring westerly wind bias.



## 4   Relationships with Other Model Biases

A weak dependence on start date suggests a weak link of the ITCZ biases documented in the previous section with SST biases, or with teleconnections with other large-scale systematic biases of the model climatology (such as in the tropical Pacific; Shonk *et al*, 2018), which are established gradually over the course of the hindcast. The similarity in the evolution of the

westerly bias over the equatorial Atlantic for different start dates (noted in Figure 5) may therefore be taken as evidence that it is primarily due to regional processes and feedbacks with little external influence. This conclusion is supported by an analysis of the association of remote biases with those in the Atlantic.

We first consider the occurrence of propagating equatorial waves as diagnosed from the wind components at 200 hPa. Both westward and eastward waves are present in the equatorial band throughout the entire period of the February hindcasts, but

there is little evidence that waves from elsewhere in the tropics are affecting the Atlantic at the time of the growth of the westerly wind bias.

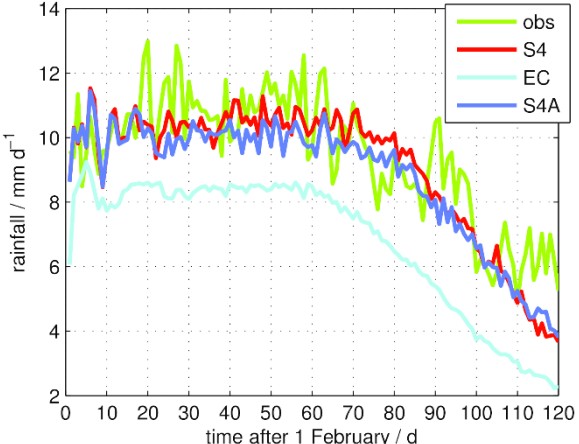

**Figure 8.  Time series of box-mean rainfall over boxes A1 and A2 (latitudes 0°—10° S; longitudes 70° W—50° W), through the first 120 days of the hindcasts initialised in February. Observed (TRMM) rainfall is shown in green; the models are shown in red, blue**

**and cyan. Full year ranges of rainfall data are used (see caption of Figure 3).**

Biases over the land masses surrounding the Atlantic could also influence the Atlantic biases (Richter *et al*, 2012). The independence of the onset of the westerly bias with start date suggests that there could be a link with an annual cycle in the local circulation patterns. The retreat of the South American Monsoon occurs typically in the middle of April (for example,

Carvalho *et al*, 2011), and it has been proposed that the springtime equatorial Atlantic westerly wind biases are associated with a deficient South American Monsoon (Richter *et al*, 2012).

We use diagnostics introduced by Raia and Cavalcanti (2008) to characterise the monsoon retreat. They used properties averaged over various boxes over land, identifying boxes A1 and A2 (spanning longitudes 70° W—50° W and latitudes 0°— 10° S; centred roughly on the Amazon basin) to show both strong monsoon rainfall properties and the heaviest annual mean

rainfall. Figure 8 shows that the observed retreat of the monsoon here occurs around day 70. Both S4 and S4A capture the



timing and the rainfall intensity of the monsoon retreat very well, while EC shows a dry bias (broadly consistent with the model's overall tendency to produce too little rainfall over land and too much over the ocean), and indicates a monsoon retreat that is about 20 days too early. Despite these differences, however, the transition to westerly bias in both coupled models occurs at the same time, suggesting that the transition is not sensitive to biases over the South American continent.

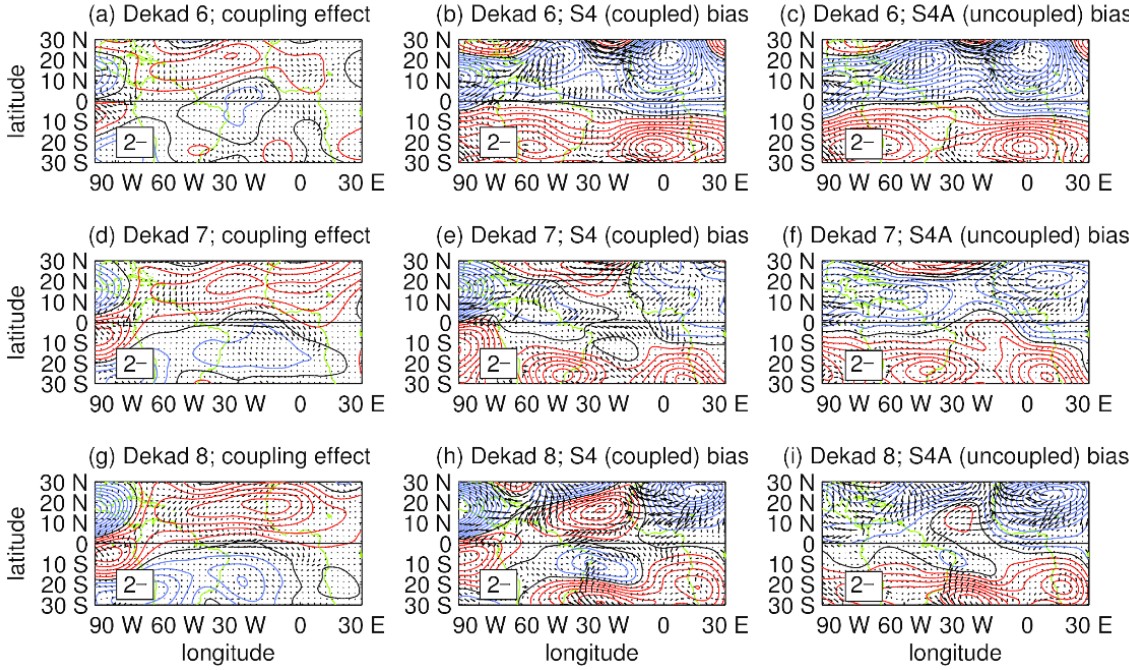

**Figure 9. Maps of dekad-mean biases in 10 m wind vector (black arrows, in m s⁻¹) and 200 hPa streamfunction (contours). The middle column shows the coupled biases in S4; the right column shows the uncoupled biases in S4A; the left column shows the coupling effect (S4 minus S4A), for dekads 6 to 8. Positive streamfunction bias contours are red; negative contours are blue and the zero contour is black. North of the equator, red contours are anticyclonic and blue contours are cyclonic. Dekads are counted from initialisation on 1 February.**

We conclude this section by examining the development of local wind biases at upper levels on a dekad-by-dekad time scale. Surface wind anomalies near the equator are often associated with vertical motion and rainfall generated by convective activity. Characteristic patterns of baroclinic motion are quickly established in response to latent heating, with a large signal aloft. Figures 9a, 9d and 9g show the coupling effect (the difference between S4 and S4A) on wind vector at 10 m and on streamfunction at 200 hPa. The first signs of the westerly bias at the surface appear in dekad 6, and grow strongly during dekad 7. The coupling effect on the streamfunction bias indicates an easterly flow aloft that develops approximately five to ten days later: in dekad 7, an easterly structure starts to develop over the equator; by dekad 8, this is well established, with a pair of anticyclones either side of the equator. This ten-day lag, combined with the baroclinic nature of the coupling effect, suggests that biases in convective heating is linking the circulation biases at the two levels, although the development of the lower-level biases in advance of the upper-level biases suggests that the biases affecting the circulation originate near the surface.



The structure of the upper-level bias over the Atlantic in S4 after dekad 7 is a distinct pair of anticyclones, one either side of the equator, resembling a Gill–Matsuno response (Matsuno, 1966; Gill, 1980) with easterlies aloft over the equator rather than westerlies (Figure 9h). This implies a cooling on the equator and hence a reduction of rainfall. The detailed circulation pattern may result from a superposition of two Gill–Matsuno patterns on either side of the equator, associated with the double ITCZ

bias. (The forcing is a band along the equator of reduced rainfall and latent heating, flanked north and south by wet bands, with the southern band wetter than the northern band; the spring panel on Figure 1 gives an idea of this distribution). A similar bias pattern appears in S4A, also developing in dekad 7, although it is markedly weaker (Figure 9i), which is consistent with the rainfall biases resembling a weaker version of a double ITCZ in the uncoupled model.

## 5    Discussion: Mechanisms of Bias Development

Our proposed chain of events for the biases in the tropical Atlantic in S4 at this stage is as follows. The model hindcasts for February and March are dominated by trade winds that are too strong, resulting in a widespread easterly bias over the ocean. Concurrently, they produce too little rainfall north of the equator in the central Atlantic, and too much rainfall in the west. The easterly wind bias is associated with a cooler ocean surface in the form of a growing cold tongue, which suppresses convection over the equator, contributing to a double ITCZ structure. At the start of April, the wind bias changes rapidly to a westerly

regime across the equatorial ocean; zonal winds reduce in magnitude to near zero. This causes the cold tongue bias to gradually disappear and eventually the equatorial Atlantic turns warm. The elevated SSTs near the equator enhance rainfall in the entire ITCZ with a strong zonal, double-banded structure, which only weakens toward the end of May to give rise to a single wide zonal band of rainfall.

Thus, the transition between the two bias regimes in S4 is associated with an increase in ITCZ rainfall that happens around the

start of April. This increase causes the erroneous southern ITCZ to grow, resulting in excess off-equatorial convergence that limits the northward reach of the southern hemisphere trade winds towards the equator. The westerly bias therefore develops in response to the formation of a strong double ITCZ with convergence south of the equator. We therefore turn our attention to the likely causes for the development of the springtime double ITCZ bias.

The classic phenomenology of the double ITCZ bias in coupled GCMs (Lin, 2007) is associated with a cold SST bias on the

equator, and we have seen that these biases do develop at least initially in the hindcasts (Figure 3). However, the later development of a double ITCZ in the absence of a cold SST bias (Figure 5c) implies that there are other processes in play. Clues can be obtained from the comparison between S4 and EC. While the distribution of rainfall between the two (northern and southern) ITCZ branches varies between the two coupled models, the overall bias pattern is similar with a dry bias north of the equator and a wet bias south of the equator that spans much of the central Atlantic. This pattern is persistent; that is, it

is found both before and after the equatorial bias regime transition. The enhancement of rainfall south of the equator and the suppression of rainfall north of the equator appear to overcome any enhancement effects from the warm SST bias found there (Figure 4).



Various earlier studies have considered the relationships between biases in rainfall over land and in circulation patterns over the Atlantic via sea-level pressure biases (for example, Chang *et al*, 2008; Richter and Xie, 2008; Richter *et al*, 2012). However, for the S4 model, we see little evidence of such influences. The zonal gradient of mean-sea-level pressure bias is found to be weak, and we have seen in Section 4 (Figure 8) that the effect of dry biases over the Amazon region is not related to those over ocean. We have also shown in Section 4 (Figure 9) that the upper-tropospheric flow patterns associated with the bias regime transition in the equatorial Atlantic does not show evidence of remote origins, but has the characteristics of a locally forced quasi-stationary Gill–Matsuno pattern.

Previous studies (cited above) focus on the long-term westerly biases that develop in spring rather than the specific development of the westerly bias regime we see in the seasonal hindcasts, and the processes that initiate the westerly bias at the start of April could be different to those that maintain it through the rest of spring. It may be difficult to isolate the mechanisms responsible for the initial development of the bias in equilibrated climate-mode integrations. Also, the development of similar biases in different models might be dependent on different mechanisms (e.g. Toniazzo and Woolnough 2014).

By contrast, the rapidly developing localised rainfall bias pattern seen in the hindcasts could be better explained by errors in the model representation of cross-equatorial flow. This is dependent on boundary layer biases; for example, according to the mechanism discussed by Pauluis (2004). If the boundary layer is sufficiently deep, the flow can cross the equator within the boundary layer. But if the boundary layer on the equator is too shallow, the flow must cross the equator in the free troposphere, which requires low-level ascent in the upwind hemisphere and descent in the downwind hemisphere. The bands of ascent and descent associated with such a low-level flow pattern would, in our case, enhance and suppress convection south and north of the equator, respectively.

During the time of the transition to the westerly bias regime, there is indeed a cross-equatorial southerly wind east of 20° W (see Figure 7), and in this region there is suppression of the northern ITCZ (downwind) and enhancement of the southern ITCZ (upwind). Furthermore, the model shows a marked reduction in meridional wind at 10 m, which is consistent with boundary layer biases forcing the low-level flow away from the surface (Figure 10). This feature starts to develop in the hindcasts at the start of April, when the southern ITCZ begins to intensify, and the extra subsidence north of the equator could be contributing to rainfall suppression in the northern ITCZ in the central and eastern Atlantic.

With the available data, we cannot draw firm conclusion with regard to boundary layer biases. A possible origin of the bias could be found upwind of the equator, where a widespread cold SST bias exists (Figure 2d). Increased stability here could be cooling the sea surface by suppressing the growth of trade-wind cumulus clouds and instead encouraging marine stratocumulus, which tends to reduce incoming solar radiation via its higher mean optical depth (for example, Schreier *et al*, 2014). Such a bias in stability could have the effect of producing a shallower boundary layer, increasing the potential for the wind to cross the equator in the free troposphere generating the rainfall biases.



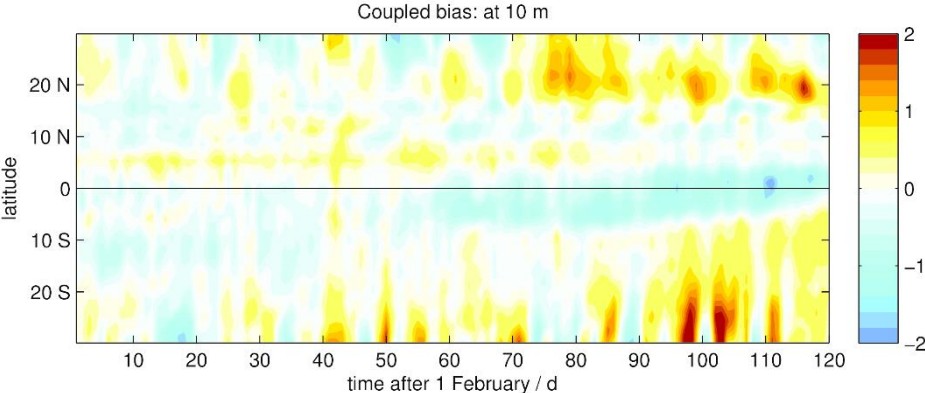

**Figure 10. Latitude–time plot showing the evolution of biases in meridional wind in S4 at 10 m (within the boundary layer), averaged over longitudes 20° W—0°. Wind speeds are in m s⁻¹ and averaged over years 1996 to 2009.**

It is interesting to note that a southern ITCZ is observed to form during the spring warming season in the Pacific and Indian Oceans. The Atlantic is thus an exception, in that no rainfall occurs over the ocean in a southern branch, and the model bias in this sector may be described as a failure to represent that exception. This raises the question alternatively as: what is it about the Atlantic that suppresses the southern ITCZ? The geometry of the land surrounding the basin might be an important factor. The Atlantic Ocean is much narrower than the other oceans and also much less meridional in structure (with western Africa

extending far out into the basin and the north-east point of Brazil extending into the south-west Atlantic). This means that the observed heating pattern is likely to be very different. Furthermore, coastally trapped waves propagating around the African coast would travel in a westward direction along the south coast of western Africa, just north of the equator. These waves could affect the rainfall patterns too, and any missing elements of coastal dynamics in the model could easily stall these waves in the Gulf of Guinea rather than allowing them to propagate westward along the coast into the central Atlantic.

**6 Summary and Conclusions**

In this study, we have used a set of seasonal hindcasts to investigate the origins of the systematic westerly wind bias that occurs on the equator in the tropical Atlantic during spring. Fully coupled initialised hindcasts from ECMWF System 4 and EC-Earth v2.3 have been used alongside a set of hindcasts from System 4 with prescribed SSTs. The use of initialised seasonal hindcasts allows us to focus on the period when the westerly wind bias first develops and attempt to identify its root cause. In the coupled

hindcasts, we found a bias regime transition from easterly wind bias to westerly wind bias that occurred rapidly at the start of April. Before the transition, in February and March, the equatorial Atlantic in the coupled models was dominated by easterly wind biases (trade winds too strong), cold SSTs and a double ITCZ, echoing similar bias patterns identified in the Pacific. Afterwards, in April and May, the trade winds reduce to near zero, leading to the reduction of the cold SST bias (which





eventually turns warm), and the merging of the two branches of the double ITCZ by the end of May. The timing of the transition is not related to start date, implying that the origins of the westerly wind bias are locked to the seasonal cycle.

Despite differences in the rainfall and SST bias patterns in the two coupled models, the timing and magnitude of the transition to westerly wind bias is very similar. The strong westerly bias, however, does not develop to the same extent when the SSTs

are prescribed, and neither does the development of a strong double ITCZ. This suggests that the presence of coupling and a double ITCZ are important factors in allowing the bias regime transition. We also found that the development of a double ITCZ is not driven solely by the presence of a cold SST bias on the equator.

The increase in rainfall observed at the start of April looks to be a factor in the growth of the westerly bias. In the coupled models, excess rainfall is spread into both branches of the double ITCZ, which results in an enhancement of the erroneous

southern ITCZ and generates an area of excess convergence south of the equator. Consequently, the southern hemisphere trade winds that should be crossing the equator and feeding a single northern ITCZ are redirected into this region, creating a calmer zone along the equator where the trade winds do not reach. Furthermore, this redirection allows the southern ITCZ to strengthen while the northern ITCZ, with its southern hemisphere moisture source cut off, weakens.

A search for possible remote origins of the Atlantic biases has returned no strong evidence. There is no clear sign of an

influence on the Atlantic from equatorial waves, and we have identified that there is no notable evidence that errors in the South American monsoon have an impact, and similarly the wind bias patterns in the upper troposphere develop in response to the surface biases rather than ahead of them.

Our main forward hypothesis is that there is an error in representation of the cross-equatorial flow that is leading to erroneous ascent to the south of the equator and descent north of the equator, leading to an enhancement of the southern ITCZ and a

reduction of the northern ITCZ. Problems with the representation of stability in the southern hemisphere (upwind of the equator) could be leading to a change of cloud regime and a possible reduction of boundary layer height, bringing about a tendency for the meridional flow in the central Atlantic to cross the equator in the free troposphere, generating the erroneous ascent and descent and corresponding impacts on rainfall. We recognise, however, that there may be other contributing factors.

## Author Contribution

Analysis of S4 and S4A data was performed mainly by JKPS and TT; analysis of EC data was performed mainly by TDD and TT. All three authors contributed to the synthesis of the results and the preparation of the manuscript.

## Competing Interests

The authors declare that they have no conflict of interest.



**Acknowledgements**

We thank Tim Stockdale, Chloé Prodhomme, Eleftheria Exarchou, Paco Doblas-Reyes, Steve Woolnough and Eric Guilyardi for insightful discussions. EC-Earth hindcast data was provided by the Barcelona Supercomputer Centre. This work was performed as part of the PREFACE project, funded by European Union Grant Agreement 603521.

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
