# Peer review of "A Double ITCZ Phenomenology of Wind Errors in the Equatorial Atlantic in Seasonal Forecasts with ECMWF Models"

_Atmospheric Chemistry and Physics, 2018_

## Referee Comment (RC1) · Anonymous Referee #1 · 10 Feb 2019

General comments

The study investigates the double-ITCZ problem in the Atlantic basin with an emphasis on wind errors. The investigation uses the hindcasts by two coupled models and an SST-forced GCM. The analysis starts with an overview of model biases and then focuses on the double-ITCZ problem during the boreal spring. After comparing model biases among the hindcasts, the study further highlights the transition of wind biases. The discussion is followed by an attempt to link the double-ITCZ problem to other biases in the hindcasts.

Overall, the study is clearly organized and provides adequate information that helps

interpret the analysis. However, the discussion of some results is incomplete and does not fully consider all the facts. These issues make some statements appear assertive or over speculative. I encourage the authors to consider the following comments and revise the manuscript accordingly.

Specific Comments

1. The attribution the double-ITCZ problem to the cross-equatorial flow is highly speculative (Line 19-21 on Page 1). Most parts of the study focus on zonal wind biases. In contrast, the cross-equatorial flow was only briefly mentioned in Section 5. Furthermore, the evidence there does not clearly suggest whether the bias of the cross-equatorial flow is a cause or a symptom, even though causality was indicated in the abstract. The ambiguity becomes more troubling if one notices that some alternative possibilities were not fully accounted for. Two apparent examples are as follows.

a) SST biases. The study discusses the potential impact of the SST biases at the equator but not those near 10N (Fig. 4). The SST biases, which develop with the easterly wind bias, may also affect the low-level wind and precipitation (e.g., Lindzen and Nigam 1987). Do they indicate any potentially important model errors? Would these biases contribute to the double-ITCZ problem (c.f. Line 5-9 on Page 12)? Why?

b) Land impact. The argument against the land impact on precipitation (Starting from Line 16 on Page 16) omitted the apparent bias along the equatorial coast between 35W and 50W (Fig. 7). While the observation has the heaviest coastal precipitation over land, the model simulations tend to have more intense precipitation offshore. Would these factors contribute to excessive oceanic precipitation in the southern hemisphere and worsen the double-ITCZ problem?

Please consider clarifying these issues and use caution when making statements.

- Lindzen, R.S. and Nigam, S., 1987. On the role of sea surface temperature gradients in forcing low-level winds and convergence in the tropics. Journal of the Atmospheric

[Figure]

Sciences, 44(17), pp.2418-2436.

2. Table 1 is helpful. But what does "Time Step" represent? The value (45 min) appears unusually large for atmospheric models with about 1-degree resolution. The table may also include the analysis periods.

3. Fig. 2a is not exactly a latitude-time plot. Please consider updating the caption.

4. Line 22 on Page 12: Could the initial easterly wind biases be related to the shock of initialization (e.g., Mulholland et al 2015; Pohlmann et al. 2017)? The problem appears common when model components are initialized separately and contain some imbalance.

- Mulholland, D.P., Laloyaux, P., Haines, K. and Balmaseda, M.A., 2015. Origin and impact of initialization shocks in coupled atmosphere–ocean forecasts. Monthly Weather Review, 143(11), pp.4631-4644. - Pohlmann, H., Kröger, J., Greatbatch, R.J. and Müller, W.A., 2017. Initialization shock in decadal hindcasts due to errors in wind stress over the tropical Pacific. Climate Dynamics, 49(7-8), pp.2685-2693.

5. Fig. 7 suggests that the S4A has severe problems in representing the zonal distribution of precipitation, even though its double-ITCZ problem is less severe. This may warrant a comment.

6. Line 16 on Page 19: It would be helpful to briefly review the mechanism. The physical reasoning that connects the boundary layer depth and the cross-equator flow is not clear here.

7. Line 8-11 on Page 16: Is there a figure that supports the statement on the equatorial waves?

8. In the last section, please consider commenting on if the findings are likely model-specific and how the findings would benefit the broader community.

Technical comments

10.5194/acp-2018-1316
2019
Atmos. Chem. Phys. Discuss.

1. Line 27 on Page 19: A missing article word in "draw firm conclusion"?

2. Many sentences appear excessively long. While they are grammatically correct, longer sentences are generally harder to follow when compared to shorter ones.

3. The title of Section 3.3 ("Focussing on the Transition Period") is not particularly informative.

4. Should the acronym (ITCZ) be explicitly defined in the title and/or the abstract?

---

## Referee Comment (RC2) · Anonymous Referee #2 · 27 Feb 2019

Summary: This study presents an attempt to quantify the bias of the ECMWF models in forecasting the equatorial westerly wind, which is caused by an unexpected double ITCZ development during the spring season. The main conclusion of this study is that this westerly wind bias is linked to an incorrect representation of the cross-equatorial meridional flows and the rainfall bias near the equator in the ECMWF models. While I could see the value of this type of work in examining different model biases for future model development and improvement, my general concern about this study is that it still lacks somewhat more in-depth diagnostic analyses that could really help identify the root of biases in the EMCWF model, which are needed for further improving the models. A conclusion such as the westerly wind bias is related to a rainfall bias is not

totally satisfying, because after all one may wonder where this rainfall bias comes from? Likewise, I do not fully see a demonstrated physical mechanism that could shed light into how the bias in the cross-equatorial flows could induce bias in the westerly wind. Having said that, I would suggest the authors to provide some additional analyses to help readers better gain more understanding into the biases in the ECMWF model as well as the physical mechanisms underlying the connection between the westerly wind bias and rainfall/cross-equatorial flow bias. Below please find my several specific concerns that highlight such a lack of analysis and/or physical explanations. These few places not exclusive, but they could at least highlight the main concern mentioned above.

Specific concerns

1. Page 5-6: Please add some possible mechanisms/analyses that explain why the wet bias in April to the south of the equator increases in both the strength and the extent later in May-June in S4 model. The authors appear to attribute the westerly wind bias to this strengthening of the moisture bias, but in the end readers would be very interested in knowing why the S4 model could develop such as moisture (i.e., wetter) bias in the first place, and how this is dynamically linked to the westerly wind bias.

2. Page 8, line 12-17: I agree that there is some correspondence between the development of a double ITCZ and the rainfall bias pattern shown in Figure 3b. However, I again do not see an explanation why this rainfall bias leads to, or at least connected to, the formation of the double ITCZ. Is the double ITCZ a manifestation of the rainfall bias, or there is indeed a physical/dynamical reason that could allow us to see how the rainfall bias accounts for this double ITCZ development? I should note that this lack of physical explanation is not only seen in this paragraph alone, but several other places as well (see, e.g., comment # 2 above). Anytime I came across this type of discussion in this work, I was hoping that I could see some more insights in the next paragraphs. However, the subsequent discussions are always shifted to showing different figures. Perhaps most readers will be left with some wondering what we actually learn from

these discussions beyond seeing some evidence in these plots.

3. It appears that there are a number of previous studies that studied the springtime westerly wind bias in GCMs and suggested that model errors in both the oceanic and atmospheric components are the cause for the westerly wind bias. I am wondering if this current study could provide a step further beyond the previous findings, i.e. specifically pinpoint where the error sources in the ECMWF models are (physics, resolution, boundary conditions, . . .). More discussions about this would be helpful for readers.

---

## Author Comment (AC1) · 17 May 2019

We have responded to the comments made by the two Reviewers in the attached PDF document.

Please also note the supplement to this comment:
https://www.atmos-chem-phys-discuss.net/acp-2018-1316/acp-2018-1316-AC1-supplement.pdf

---

## Author Response (AR1)

**Responses to Reviewers' Comments on: *A Double ITCZ Phenomenology of Wind Errors in the Equatorial Atlantic in Seasonal Forecasts with ECMWF Models**

Please find below our responses to the comments from the reviewers on our manuscript. For clarity, we have copied their reviews into this document and added our responses in blue text. Page and line numbers in our responses correspond to the resubmitted version. To make it easier for the reviewers, updated parts of the manuscript have been highlighted in yellow.

**REVIEWER "A"**

General comments
The study investigates the double-ITCZ problem in the Atlantic basin with an emphasis on wind errors. The investigation uses the hindcasts by two coupled models and an SST-forced GCM. The analysis starts with an overview of model biases and then focuses on the double-ITCZ problem during the boreal spring. After comparing model biases among the hindcasts, the study further highlights the transition of wind biases. The discussion is followed by an attempt to link the double-ITCZ problem to other biases in the hindcasts. Overall, the study is clearly organized and provides adequate information that helps interpret the analysis. However, the discussion of some results is incomplete and does not fully consider all the facts. These issues make some statements appear assertive or over speculative. I encourage the authors to consider the following comments and revise the manuscript accordingly.
Specific Comments

1.  The attribution the double-ITCZ problem to the cross-equatorial flow is highly speculative (Line 19-21 on Page 1). Most parts of the study focus on zonal wind biases. In contrast, the cross-equatorial flow was only briefly mentioned in Section 5. Furthermore, the evidence there does not clearly suggest whether the bias of the cross-equatorial flow is a cause or a symptom, even though causality was indicated in the abstract.
    Indeed, the impact of the cross-equatorial flow bias is a speculative contribution to the bias. We have edited the end of the abstract to remove the implication of causality. We have also reworded it following a short piece of extra analysis (see response to comment 6 below).
    The ambiguity becomes more troubling if one notices that some alternative possibilities were not fully accounted for. Two apparent examples are as follows.

a)  SST biases. The study discusses the potential impact of the SST biases at the equator but not those near 10N (Fig. 4). The SST biases, which develop with the easterly wind bias, may also affect the low-level wind and precipitation (e.g., Lindzen and Nigam 1987). Do they indicate any potentially important model errors? Would these biases contribute to the double-ITCZ problem (c.f. Line 5-9 on Page 12)? Why?
    The off-equatorial SST bias pattern could be contributing to the development of a double ITCZ via changes to the circulation -- we do see a meridional component to the wind bias through February, March and April, which would be consistent (Figure 3b). Indeed, it could

partly explain why there is a double ITCZ pattern in the rainfall in the April hindcasts despite the absence of a cold bias on the equator (Figure 5c). However, the detail of the SST and rainfall bias patterns suggests that this is not the only factor.

Lindzen and Nigam's model is not applicable over the Equator, and its prediction of the zonal wind component in particular is inaccurate (as pointed out by the authors). For example, it fails to account for total vorticity conservation. In the S4 forecasts, we do see a surface wind divergence field associated with the zonally asymmetric component of SST biases at 10N, qualitatively consistent with Lindzen and Nigam. However, the effect is small, and the leading-order wind and precipitation errors are nearly identical between the coupled forecasts and those using prescribed SSTs. In particular, the convergence south of the Equator in the west, and thus the double-ITCZ character of the precipitation bias field, is not associated with an SST error pattern initially. We therefore regard the northward shift of the ITCZ (mostly in the west; cf Figure 1b) as a contributing, but not causative or leading-order factor in the bias development.

We have added text to Section 5 highlighting these points (page 18, line 30 onwards).

b) Land impact. The argument against the land impact on precipitation (Starting from Line 16 on Page 16) omitted the apparent bias along the equatorial coast between 35W and 50W (Fig. 7). While the observation has the heaviest coastal precipitation over land, the model simulations tend to have more intense precipitation offshore. Would these factors contribute to excessive oceanic precipitation in the southern hemisphere and worsen the double-ITCZ problem?

We have considered this hypothesis very carefully in the initial stages of our analysis, but eventually we had to discard it as a possible leading-order explanation. The Atlantic biases in these three models consist of two bias patterns: a double ITCZ and the bias pattern on the east coast of South America (dry over land, wet over ocean). The magnitude of the latter is shown best in the S4A hindcasts, where the coastal bias pattern dominates the overall bias distribution. Given the location of the southern branch of the ITCZ, it is very likely that this bias enhances the southern ITCZ, and may be important in the development of the overall bias pattern in the coupled models.

However, we could not find any phenomenological connection between this coastal bias and the overall evolution of the Atlantic ITCZ, either temporally, or between different models. The reader may see this by comparing Figure 3 and Figure 7. S4A has a stronger coastal bias than S4, but much weaker marine ITCZ and westerly wind biases south of the equator. EC has a weaker coastal bias, but a similar marine double ITCZ and westerly wind bias. The marine ITCZ evolve with remarkable synchronicity among the two coupled models, while the coastal evolution is very different. We came to the conclusion that the coastal bias is a bit of a red herring, in that is may represent important local biases, but only a small effective displacement of precipitation in a basin-wide view.

Our second, and separate consideration regarding the impact of land precipitation, which is addressed by the reviewer here, is in the context of large-scale land rainfall biases over vast areas of Africa and South America rather than local coastal biases. We found little evidence of such large-scale patterns driving the biases (as shown by the lack of rainfall biases during the monsoon retreat -- Figure 8).

The contribution to the southern ITCZ of the coastal bias pattern is mentioned in section 3.3 (page 13, line 17 onwards). We emphasize that the biases discussed in Section 4 are large-scale biases over much greater areas (page 16, line 18).

Please consider clarifying these issues and use caution when making statements.

Lindzen, R.S. and Nigam, S., 1987. On the role of sea surface temperature gradients in forcing low-level winds and convergence in the tropics. Journal of the Atmospheric Sciences, 44(17), pp.2418-2436.

2. Table 1 is helpful. But what does "Time Step" represent? The value (45 min) appears unusually large for atmospheric models with about 1-degree resolution. The table may also include the analysis periods.

"Time step" is the increment of time that the model advances by in one cycle of calculations. Various individual processes (e.g. dynamic adjustment, some elements of convection) are sub-stepped, but 45 minutes is indeed the discrete time increment used e.g. to advect tracers and to advance the model state vector (cf also Molteni et al., 2011). We have reworded this to be "model time step" in Table 1 (page 4). We have also included the analysis periods as requested.

3. Fig. 2a is not exactly a latitude-time plot. Please consider updating the caption.

We have reworded the caption to make this clearer (page 5, line 7). We have also updated Figure 2 so that the black spots marking insignificance on panel (a) appear for the second occurrence of January and February (these were missing previously).

4. Line 22 on Page 12: Could the initial easterly wind biases be related to the shock of initialization (e.g., Mulholland et al 2015; Pohlmann et al. 2017)? The problem appears common when model components are initialized separately and contain some imbalance.

- Mulholland, D.P., Laloyaux, P., Haines, K. and Balmaseda, M.A., 2015. Origin and impact of initialization shocks in coupled atmosphere–ocean forecasts. Monthly Weather Review, 143(11), pp.4631-4644.
- Pohlmann, H., Kröger, J., Greatbatch, R.J. and Müller, W.A., 2017. Initialization shock in decadal hindcasts due to errors in wind stress over the tropical Pacific. Climate Dynamics, 49(7-8), pp.2685-2693.

Shock is potentially a problem for both coupled models. Both S4 and EC were initialised with ERA-Interim and ORA-S4. The atmosphere component of EC was used to generate ERA-Interim and the ocean component of S4 was used to generate ORA-S4. So both models have potential sources of shock, which can occur even across different versions of the same model -- see Mulholland et al (2015).

We see evidence of model shock in the SST bias field. Biases in SST from day 1 in S4 are positive and about 0.3 °C, while biases in EC start from near zero. Mulholland et al showed that imbalances through differences in model version could produce day 1 biases of order 0.5 °C (see their Figure 9), which is consistent. Further evidence for these early SST biases being a shock come from Figure 2, which shows that SST biases for the same period at longer lead times are cold rather than warm (at least south of the equator). We have added a short paragraph on the possible evidence of model shock in SSTs (page 10, line 19).

It is less clear whether the initial easterly wind biases could be associated with shock. Given that the incompatibility in each of EC and S4 are in different model components, the similarity of their initial wind bias development despite this suggests not. However, Shonk et al (2018) found that easterly wind biases in the western Pacific that were indicative of the

occurrence of shock, and tended to be short-lived, lasting for a month or two. In contrast, the Atlantic easterly biases are much more persistent.

5. Fig. 7 suggests that the S4A has severe problems in representing the zonal distribution of precipitation, even though its double-ITCZ problem is less severe. This may warrant a comment.
Both S4 and S4A have problems with their rainfall being situated too much to the west, although it is more pronounced in the absence of a strong double ITCZ structure in S4A. We have commented on this alongside the discussion of the land--ocean bias pattern as described in comment 1b above (page 13, line 17). What might be considered most remarkable with S4A is the lack of equally severe (easterly) wind biases in the presence of that zonal precipitation bias. It is an indication of compensating errors in the simulated winds. The zonal asymmetry in precipitation should be expected to produce an equatorial easterly error, and to generally impede cross-equatorial flow (Rodwell and Hoskins 1995). This might be compensated by a strong pressure gradient in a sufficiently thick marine PBL (Dvorkin and Paldor 1999; Pauluis 2004). We surmise that the latter falls away in the coupled simulations, leading to convergence south of the Equator, the formation of a double ITCZ, and westerly wind anomalies.

6. Line 16 on Page 19: It would be helpful to briefly review the mechanism. The physical reasoning that connects the boundary layer depth and the cross-equator flow is not clear here.
Extra detail has been added on the cross-equatorial flow (page 19, line 20). The underlying mechanism is the dynamical barrier represented by the zero contour line of total vorticity or PV (Rodwell and Hoskins 1995). Flow along or between isoentropes that do intersect the ground cannot cross this line. The forces governing this flow must thus reside within the PBL. We refer to three paper that discuss this problem in depth: Rodwell and Hoskins (1995), Dvorkin and Paldor (1999), and Pauluis (2004). In all of these the importance of pressure gradients due to gradients in PBL thickness is noted. The latter two in particular show a threshold behaviour, disallowing cross-equatorial flow below a certain PBL thickness. In such conditions, on the one hand, the PBL shows a line convergence as the Equator is approached (Pauluis 2004); on the other hand PV must be generated diabatically within the PBL to maintain the flow (Rodwell and Hoskins 1995). We surmise that this is when a secondary ITCZ branch forms. As the reviewer notes, this argument is speculative and does not lead to a firm conclusion of this work, but rather to a lead for future work. Nevertheless we can say that the differences in the climatologies of S4 and S4A are consistent with this argument (while inconsistent with every other one we could think of). We attach a figure showing the 700hPa-1000hPa thickness for S4 and S4A in February, March and April, and their differences (Figure A, below). We also show the differences in lower-tropospheric stability (LTS) for these months (Figure B). LTS differences do not change much, indicating that the thickness is a good proxy for PBL depth (with a minus sign). The reviewer will note that in March and April S4 has a reduced thickness gradient just south of the Equator compared with S4A. According to the argument above, and assuming that a threshold is crossed, this will tend to generate meridional convergence. For a while, this may be compensated by zonal divergence in the easterlies, but if and when it results in a sufficiently

thick and moist PBL south of the Equator, convection may be triggered, allowing to initiate the cross-equatorial flow that eventually feeds the African Monsoon by means of a southern ITCZ branch. We have clarified this in Section 5 (page 20, line 7) and included a subset of panels from Figure A in this response into the paper as Figure 11. The conclusions have been accordingly modified (page 22, line 15), as has the end of the Abstract.

7. Line 8-11 on Page 16: Is there a figure that supports the statement on the equatorial waves? Given the number of figures already included in the paper, we feel it unnecessary to add a figure showing that there is no notable wave pattern. We have clarified that a figure for this is not shown (page 16, line 10).

8. In the last section, please consider commenting on if the findings are likely model-specific and how the findings would benefit the broader community.
We have added a paragraph on this to the end of section 6 (page 22, line 28 onwards). This paragraph states that the representation of the trade-cumulus sub-equatorial PBL is likely a crucial aspect of the seasonal transition from equinoctial to solstitial circulation and that model development efforts should focus there.  Previous studies have shown that, while some models may have similar bias patterns, there can be differences in the origins of these (see, for example, Toniazzo and Woolnough, 2014 or Vannière et al, 2013). This is perhaps further highlighted by the fact that we found our bias origins to be quite different to those reported by other studies (such as Richter et al, 2012, as mentioned in Section 5). In other words, it could be that these results are quite specific to the ECMWF models. But the results could be used to inform a variety of future studies, including model comparisons. Our paper also benefits the community by increasing the number of citations.

Technical comments
9. Line 27 on Page 19: A missing article word in "draw firm conclusion"?
This has been changed to "draw a firm conclusion" (page 20, line 14).

10. Many sentences appear excessively long. While they are grammatically correct, longer sentences are generally harder to follow when compared to shorter ones.
We have broken up a few instances of long sentences.

11. The title of Section 3.3 ("Focussing on the Transition Period") is not particularly informative. We have reworded the title, replacing "Transition Period" with "Onset of the Westerly Wind Bias".

12. Should the acronym (ITCZ) be explicitly defined in the title and/or the abstract?
We have expanded the acronym in the Abstract. We will keep it as "ITCZ" in the title though for brevity.

**REVIEWER "B"**

Summary: This study presents an attempt to quantify the bias of the ECMWF models in forecasting the equatorial westerly wind, which is caused by an unexpected double ITCZ development during the spring season. The main conclusion of this study is that this westerly wind bias is linked to an incorrect representation of the cross-equatorial meridional flows and the rainfall bias near the equator in the ECMWF models. While I could see the value of this type of work in examining different model biases for future model development and improvement, my general concern about this study is that it still lacks somewhat more in-depth diagnostic analyses that could really help identify the root of biases in the ECMWF model, which are needed for further improving the models. A conclusion such as the westerly wind bias is related to a rainfall bias is not totally satisfying, because after all one may wonder where this rainfall bias comes from?

The aim of this paper was to try to understand the origins of biases in the ECMWF models in the tropical Atlantic and, with the time and data available to us, we have made as much progress as we could. Along the way, we have been careful not to oversell the paper and promise too much (see the motivation sentences at the bottom of page 2/top of page 3). However, we do have a conclusion from our analysis, even if it relies to a great extent on negative results. That conclusion is that upstream PBL state errors are at the origin of both rainfall and wind bias, and that they are linked with the representation of the marine trade-cumulus PBL. Please see also our response to Reviewer A.

Likewise, I do not fully see a demonstrated physical mechanism that could shed light into how the bias in the cross-equatorial flows could induce bias in the westerly wind. Having said that, I would suggest the authors to provide some additional analyses to help readers better gain more understanding into the biases in the ECMWF model as well as the physical mechanisms underlying the connection between the westerly wind bias and rainfall/cross-equatorial flow bias.

The hypothesis of cross-equatorial flow question is one that would be interesting to follow but, with present limitations of time and data, this is not possible. This is why we have left the cross-equatorial flow idea as a discussion point and as a lead for further work..

Below please find my several specific concerns that highlight such a lack of analysis and/or physical explanations. These few places not exclusive, but they could at least highlight the main concern mentioned above.

Specific concerns

1. Page 5-6: Please add some possible mechanisms/analyses that explain why the wet bias in April to the south of the equator increases in both the strength and the extent later in May-June in S4 model. The authors appear to attribute the westerly wind bias to this strengthening of the moisture bias, but in the end readers would be very interested in knowing why the S4 model could develop such as moisture (i.e., wetter) bias in the first place, and how this is dynamically linked to the westerly wind bias.

The main factor affecting the growth of the wet bias to the south through May and June is most likely the transition of the SST bias on the equator from cold to warm, and the corresponding transition from suppression to enhancement of equatorial convection. We have flagged this up in the discussion at the end of Section 3.2 (page 12, line 29).

2. Page 8, line 12-17: I agree that there is some correspondence between the development of a double ITCZ and the rainfall bias pattern shown in Figure 3b.

> However, I again do not see an explanation why this rainfall bias leads to, or at least connected to, the formation of the double ITCZ. Is the double ITCZ a manifestation of the rainfall bias, or there is indeed a physical/dynamical reason that could allow us to see how the rainfall bias accounts for this double ITCZ development?

The rain represented in Figure 3b is the actual value according to the model, not the bias (see caption). In our view, the double ITCZ is a manifestation of the rainfall bias. That is, a double ITCZ requires a rainfall bias to exist, although a rainfall bias need not be a double ITCZ. In this study, we are looking into the rainfall and wind biases, which our analysis shows to be a double ITCZ problem. We have reworded the start of the paragraph suggested to make it clearer what we mean (page 8, line 12).

> I should note that this lack of physical explanation is not only seen in this paragraph alone, but several other places as well (see, e.g., comment # 2 above). Anytime I came across this type of discussion in this work, I was hoping that I could see some more insights in the next paragraphs. However, the subsequent discussions are always shifted to showing different figures. Perhaps most readers will be left with some wondering what we actually learn from these discussions beyond seeing some evidence in these plots.

This paper is structured around testing a series of hypotheses for the formation of the April biases. Our efforts have mostly led to negative results. Thus, beyond the initial description of the bias itself, much of the evidence presented simply serves to discard one hypothesis or another. This is less satisfying that really being able to explain something, but we felt it is important to publish as many of these hypotheses are often (sometimes unthinkingly) assumed valid in other work. We do end the paper with one hypothesis that we cannot discard, which is based on the dynamical constraints of the flow in the marine PBL.
The tendency of the discussions to dash from figure to figure without much insight is a feature of the way that we have structured the paper -- particularly Section 3, which runs through the figures, extracts the evidence, then sums up the insights at the end. We have added "signposts" into the text at various points (mostly in Section 3.2) to point out what we are hoping to discover from each figure, then extra clarification at the end of Section 3.2 of when the insights begin (page 12, line 16). We have also heavily edited the paragraph at the end of Section 3.1 to sum up the insights obtained so far, and moved the introduction of the concept of the two "bias regimes" here (page 7, line 9).

3. It appears that there are a number of previous studies that studied the springtime westerly wind bias in GCMs and suggested that model errors in both the oceanic and atmospheric components are the cause for the westerly wind bias. I am wondering if this current study could provide a step further beyond the previous findings, i.e. specifically pinpoint where the error sources in the ECMWF models are (physics, resolution, boundary conditions, . . .). More discussions about this would be helpful for readers.

We acknowledge this important point. The hindcast data available for the ocean was too limited, and verification is more problematic, so our analysis is limited to the atmosphere. We must therefore allow the possibility that underlying ocean conditions lead to the double ITCZ problem in both models. Even so, the rapidity of the atmospheric regime change in April almost regardless of the underlying SSTs, and its insensitivity to hindcast initialisation time, suggests that the controlling mechanism indeed resides in the atmosphere. With the data

that we have, going far beyond what we have presented here, and what other authors have presented in the past, is not possible. All of the points raised in this comment could motivate future work, though, that would be very useful in understanding model biases and has been added at the end of the conclusions section (page 22, line 22).

[Figure]

**Figure A.** Maps of monthly mean geopotential thickness (in m) between 700 hPa and 1,000 hPa. The left column shows the bias in S4 with respect to ERA-Interim; the middle column shows the bias in S4A; the right column shows the effect of coupling on stability (S4 minus S4A). Averaged over all 14 years of hindcast.

[Figure]

**Figure B.** As Figure A, but showing lower tropospheric stability differences (in °C), determined the difference between potential temperature at 700 hPa and 1,000 hPa.